# A clinical prediction model to differentiate tuberculous spondylodiscitis from pyogenic spontaneous spondylodiscitis

Thamrong Lertudomphonwanit[1], Chirtwut Somboonprasert[1,2], Kittiphon Lilakhunakon[1,3], Suphaneewan Jaovisidha[4], Thumanoon Ruangchaijatuporn[4], Praman Fuangfa[4], Sasivimol Rattanasiri[5], Siriorn Watcharananan[6], Pongsthorn Chanplakorn[1]*

1 Faculty of Medicine Ramathibodi Hospital, Department of Orthopaedics, Mahidol University, Bangkok, Thailand, 2 Sawangdaendin Crown Prince Hospital, Sakon Nakhon, Thailand, 3 Roiet Hospital, Roiet, Thailand, 4 Faculty of Medicine Ramathibodi Hospital, Department of Diagnostic and Therapeutic Radiology, Mahidol University, Bangkok, Thailand, 5 Faculty of Medicine, Ramathibodi Hospital, Department of Clinical Epidemiology and Biostatistics, Mahidol University, Bangkok, Thailand, 6 Faculty of Medicine Ramathibodi Hospital, Departments of Medicine, Mahidol University, Bangkok, Thailand

* pongsthornc@gmail.com

## Abstract

### Background

Microbiological diagnosis of tuberculous spondylodiscitis (TS) and pyogenic spontaneous spondylodiscitis (PS) is sometime difficult. This study aimed to identify the predictive factors for differentiating TS from PS using clinical characteristics, radiologic findings, and biomarkers, and to develop scoring system by using predictive factors to stratify the probability of TS.

### Methods

A retrospective single-center study. Demographics, clinical characteristics, laboratory findings and radiographic findings of patients, confirmed causative pathogens of PS or TS, were assessed for independent factors that associated with TS. The coefficients and odds ratio (OR) of the final model were estimated and used to construct the scoring scheme to identify patients with TS.

### Results

There were 73 patients (51.8%) with TS and 68 patients (48.2%) with PS. TS was more frequently associated with younger age, history of tuberculous infection, longer duration of symptoms, no fever, thoracic spine involvement, $\geq 3$ vertebrae involvement, presence of paraspinal abscess in magnetic-resonance-image (MRI), well-defined thin wall abscess, anterior subligamentous abscess, and lower biomarker levels included white blood cell (WBC) counts, erythrocyte-sedimentation-rate (ESR), neutrophil fraction, and C-reactive protein (all p < 0.05). Multivariate logistic regression analysis revealed significant predictors of TS included WBC $\leq 9,700/mm^3$ (odds ratio [OR] 13.11, 95% confidence interval [CI] 4.23–40.61), neutrophil fraction $\leq 78\%$ (OR 4.93, 95% CI 1.59–15.30), ESR $\leq 92$ mm/hr

**Data Availability Statement:** All relevant data are within the manuscript and its Supporting Information files.

**Funding:** The author(s) received no specific funding for this work.

**Competing interests:** The authors have declared that no competing interests exist.

(OR 4.07, 95% CI 1.24–13.36) and presence of paraspinal abscess in MRI (OR 10.25, 95% CI 3.17–33.13), with an area under the curve of 0.921. The scoring system stratified the probability of TS into three categories: low, moderate, and high with a TS prevalence of 8.1%, 29.6%, and 82.2%, respectively.

## Conclusions

This prediction model incorporating WBC, neutrophil fraction counts, ESR and presence of paraspinal abscess accurately predicted the causative pathogens. The scoring scheme with combination of these biomarkers and radiologic features can be useful to differentiate TS from PS.

## Introduction

Spontaneous infectious spondylodiscitis (SIS) is an infection of the spine. The global incidence of SIS ranges from 8 to 24 cases/million inhabitants/year [1,2]. As a potentially life-threatening disease, SIS has mortality rate of 3%–20% [3–6]. Because of increases in susceptible populations and improved diagnostic modalities, the incidence of SIS has increased [7–9]. Pyogenic spondylodiscitis (PS) and tuberculous spondylodiscitis (TS) are common causes of SIS, accounting for 40%–80% and 17%–40% of all SIS cases, respectively [1,3,4,10,11]. Delayed diagnosis and treatment due to low specific signs and symptoms may lead to catastrophic outcomes [12,13].

Differentiating TS from PS is essential for appropriate management. However, diagnosing these two entities is challenging because of nonspecific signs and symptoms. Microbiological diagnosis is the benchmark for differentiating TS from PS, but determination of microorganism is difficult. Previous reports revealed 10%–30% negative culture rate in PS patients [14,15], TS positive-culture rate yields between 50% and 70% and requires 3 weeks to obtain result [16,17]. In the presence of the impossibility of microbiological diagnosis; clinical, radiological findings and biomarkers may help to determine potential etiologic microorganism. Previous studies have distinguished radiological findings between TS and PS [18,19]. Though, only radiological findings without clinical characteristics and biomarkers are inadequate for differentiating [13]. Limited data also exists regarding the biomarker cutoff values for differentiating between TS and PS [11].

To the best of our knowledge, there is currently no scoring system for diagnosing TS. The present study thus aimed to (1) identify the predictive factors for differentiating TS from PS using clinical characteristics, radiological findings, and biomarkers, and to (2) develop a scoring system by using the predictive factors to stratify the probability of TS in SIS patients.

## Materials and methods

### Study design and patient characteristics

This single-institution retrospective comparative study included patients with a diagnosis of SIS between January 2008 and December 2021 (the first phase included patients from January 2008 to December 2016 and the extended phase included patients from January 2017 to December 2021, using the same study protocol). The human research ethics committee of Faculty of Medicine, Ramathibodi Hospital, Mahidol University was reviewed and approved research protocol in both cohorts, the ethical clearance certificates were as follows; MURA

2015/279 for cohort 2008–2016 and MURA 2022/173 for extended cohort 2017–2021. As this study used secondary data and data analyzing was performed anonymously, therefore, the informed consents process was waived by the IRB. Our hospital is 1300-bed, University hospital with government support. Reimbursement systems at our hospital are covered by multiple sources such as cash, universal coverage by government, workers' insurance, government pay for government personnel, and private insurance. The medical records of the adult patients (age ≥ 15 years) diagnosed with the first episode of either PS or TS and followed to the end of treatment were reviewed. PS was diagnosed based on one of following criteria: (1) causative organism was isolated from spinal tissue at the suspected site of infection or (2) the patient had compatible spondylodiscitis signs and symptoms with radiological evidence of vertebral infection and positive blood cultures. TS was diagnosed based on one of following criteria: (1) histopathological findings established granuloma formation or positive polymerase chain reaction (PCR) for mycobacterium tuberculosis (M-TB) or (2) spinal tissue culture returned positive for M-TB. The exclusion criteria were (1) spinal infection associated with prior surgical or interventional procedure, (2) lacked of a definitive diagnosis (3) incomplete data, such as missing laboratory or radiological findings and (4) patients with simultaneous PS and TS (Fig 1).

## Data collection

Data collected were demographic characteristics, predisposing factors, and clinical manifestations (e.g., patient history, chief complaints, history of fever, underlying diseases, and time elapsed to diagnosis). History suggestive of bacteremia e.g., fever and chill or sepsis, previous history of tuberculosis or ongoing infection, and previous history of surgery and/or spinal surgery were also recorded, as were hospital stays and in-hospital mortality. Laboratory data included white blood cell (WBC) count, neutrophil fraction, erythrocyte sedimentation rate (ESR), C-reactive protein (CRP), and alkaline phosphatase level (ALP). Radiographic findings included level and number of spinal vertebrae involved. Magnetic resonance imaging (MRI) findings included the presence of paraspinal/epidural abscesses and vertebral, disc, and posterior element involvement. The X-rays and MRIs were independently evaluated by two expert musculoskeletal radiologists who were unaware of definitive diagnosis. If findings were discordant, the decision were made by consensus and reviewed by senior expert musculoskeletal radiologist.

## Statistical analysis

Mean (SD) and median (range) were used to describe continuous data, and frequency (percentage) was used for categorical data. Independent t test (or quantile regression) and chi-square test (or exact test) were used to compare factors between TS and PS groups for continuous and categorical data, respectively.

The receiver operating characteristic (ROC) and Youden index analysis were used to determine the optimal biomarker cutoff values for differentiating TS from PS. Simple logistic regression was used to screen factors associated with TS. Factors with a p value of ≤0.1 were considered simultaneously in the multivariate logistic model. Forward selection was used for model selection. The performances of the final model were assessed by Hosmer-Lemeshow goodness-of-fit statistic, and the area under the ROC, known as the concordance (C) statistic. The ROC analysis was also used to estimate sensitivity, specificity, and positive and negative predictive values with a threshold of 0.05 of the final model.

The coefficients and odds ratio (OR) of the final model were estimated. The rounded up estimated OR was used to construct the scoring scheme. Individual patients were allocated scores according to each factor, and those scores were then summed up with constant value as

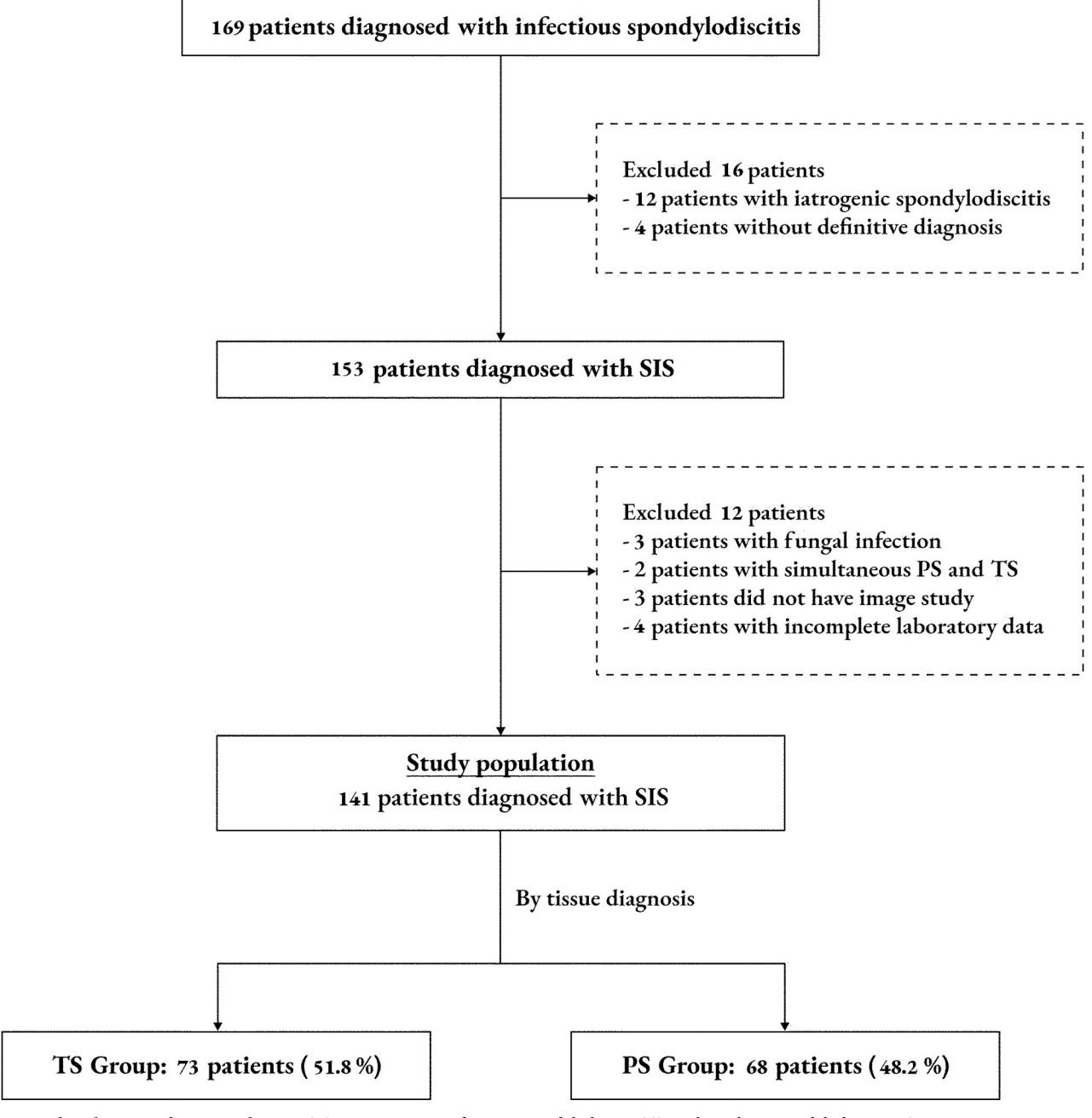

**Fig 1. Flow diagram of patient selection.** SIS = spontaneous infectious spondylodiscitis, TS = tuberculous spondylodiscitis, PS = pyogenic spondylodiscitis. Iatrogenic spondylodiscitis was defined as spinal infection associated with prior surgical or interventional procedure that entered the spinal area.

total scores. Score performances—that is, the sensitivity, specificity, and positive and negative likelihood ratios ($LR^+/LR^-$)—were calculated according to each possible total score. The total scores were then classified as low, moderate, and high risk of TS based on the strength of the $LR^+$. Statistical analyses were performed using Stata 16 software (Stata Statistical Software: Release 16, 2019; StataCorp LP, College Station, TX, USA).

## Results

### Patient characteristic and demographic data

During study period, 141 patients treated for PS or TS at our institution who met the criteria were enrolled in the study. Among 68 patients with PS, causative organism was isolated by spinal tissue cultures in 51 (75%) cases and by blood cultures in 35 (51.5%) cases. The causative organisms in PS group were other Streptococcus spp. (21; 30.9%), Staphylococcus aureus (18; 26.5%), gram-negative bacteria (17; 25.0%) and Streptococcus group B (12; 17.6%) The same organism was identified in both blood and tissue cultures in 18 (26.5%) cases. Among 73 patients with TS, M-TB was isolated from spinal tissue cultures in 49 (67.1%) cases, 16 (21.9%) were confirmed by positive PCR, and 8 (11%) were diagnosed by positive finding for granuloma formation. The baseline demographics are summarized in Table 1. Mean age at time of diagnosis was 58.8 years (range17–92; median 60 years); 57.4% of patients were female. Mean age of patients in TS group was less than in PS group (55.8 years vs. 62.0 years; p = 0.022). The proportion of patients with history suggestive of bacteremia was significantly higher in PS group than in TS group (64.7% vs. 0%; p < 0.001). Active tuberculosis was more frequently associated with TS than with PS (43.8% vs. 1.5%; p < 0.001). The proportion of patients with diabetes was significantly higher in PS group than in TS group (29.4% vs. 15.1%; p = 0.040) (Table 1).

### Clinical presentations and laboratory data

The median diagnostic delay was longer in TS group compared to PS group (90 vs. 21 days; p < 0.001). Back pain was the most common symptom in both groups. Fever exceeding 38.0˚C was less frequent in the TS group (15/73; 20.5%) compared to with that in the PS group

**Table 1. Patient demographics.**

| Variables | All (n = 141) | TS (n = 73) | PS (n = 68) | P value |
|---|---|---|---|---|
| Age* | 58.8 ± 16.3 | 55.8 ± 18.7 | 62.0 ± 12.6 | 0.022[‡] |
| Female sex[†] | 81 (57.4) | 44 (60.3) | 37 (54.4) | 0.482 |
| Bacteremia[†] | 44 (31.2) | 0 (0.0) | 44 (64.7) | < 0.001[‡] |
| Exposure to immunosuppressive drugs[†] | 6 (4.3) | 4 (5.5) | 2 (2.9) | 0.878 |
| Active tuberculosis other than spine[†] | 33 (23.4) | 32 (43.8) | 1 (1.5) | < 0.001[‡] |
| **Comorbidity, n (%)** | | | | |
| Diabetes[†] | 31 (22.0) | 11 (15.1) | 20 (29.4) | 0.040[‡] |
| Hypertension[†] | 54 (38.3) | 25 (34.2) | 29 (42.6) | 0.305 |
| Dyslipidemia[†] | 42 (29.8) | 22 (30.1) | 20 (29.4) | 0.925 |
| Chronic kidney disease[†] | 13 (9.2) | 8 (11.0) | 5 (7.4) | 0.460 |
| Chronic liver disease[†] | 8 (5.7) | 2 (2.7) | 6 (8.8) | 0.155 |
| Cardiovascular disease[†] | 8 (5.7) | 3 (4.1) | 5 (7.4) | 0.320 |
| Cerebrovascular disease[†] | 4 (2.8) | 3 (4.1) | 1 (1.5) | 0.621 |
| Malignancy[†] | 7 (5.0) | 4 (5.5) | 3 (4.4) | 0.749 |
| Connective tissue disease[†] | 3 (2.1) | 1 (1.4) | 2 (3.0) | 0.607 |
| HIV infection[†] | 2 (1.4) | 2 (2.7) | 0 (0.0) | 0.497 |

TS = Tuberculous spondylodiscitis, PS = pyogenic spondylodiscitis, HIV = human immunodeficiency virus.

*The values are given as mean and standard deviation.

[†]The values are given as the number of patients, with the percentage in parentheses.

[‡]Significant (p < 0.05).

**Table 2. Clinical characteristics and laboratory findings in patients with tuberculous and pyogenic spondylodiscitis.**

| Variables | All (n = 141) | TS (n = 73) | PS (n = 68) | P value |
|---|---|---|---|---|
| **Clinical manifestations, n (%)** | | | | |
| Time elapsed to diagnosis¶ (days) | 60.0 (1, 1095) | 90.0 (5, 1,095) | 21.0 (1, 730) | <0.001‡ |
| Back pain† | 134 (95.0) | 69 (94.5) | 65 (95.6) | 1.000 |
| Fever† | 55 (39.0) | 15 (20.5) | 40 (58.8) | <0.001‡ |
| Neurological deficits† | 60 (42.6) | 34 (46.6) | 26 (38.2) | 0.317 |
| Cauda equina syndrome† | 24 (17.0) | 11 (15.1) | 13 (19.1) | 0.523 |
| **Laboratory findings** | | | | |
| WBC Count* (per mm$^3$) | 11,155 ± 5682 | 8,159 ± 3324 | 14371 ± 5943 | < 0.001‡ |
| Neutrophil count* (%) | 73.5 ± 11.9 | 68.6 ± 10.2 | 78.8 ± 11.3 | < 0.001‡ |
| ESR* (mm/h) | 73.7 ± 28.0 | 61.6 ± 25.8 | 86.8 ± 24.1 | < 0.001‡ |
| CRP level¶ (mg/L) | 42.0 (1.1, 302) | 24.4 (1.7, 250.0) | 84.8 (1.1, 301.6) | < 0.001‡ |
| ALP level¶ (IU/L) | 110 (45.0, 784.0) | 107 (45.0, 784.0) | 126.0 (56.0, 436.0) | 0.134 |

TS = Tuberculous spondylodiscitis, PS = pyogenic spondylodiscitis, WBC = white blood cell, ESR = erythrocyte sedimentation rate, CRP = C-reactive protein, ALP = alkaline phosphatase.

*The values are given as mean and standard deviation.

†The values are given as the number of patients, with the percentage in parentheses.

¶The values are given as median, with the range in parenthesis.

‡Significant (p < 0.05).

(40/68; 58.8%) (p < 0.001). There were no statistical differences in the frequency of back pain or neurological deficits including bowel and bladder involvement between groups (Table 2).

The laboratory data are summarized in Table 2. Compared with the PS group, the TS group had lower mean or median values in the following parameters: WBC count (8,159 vs. 14,371/mm$^3$; p < 0.001), neutrophil count (68.6% vs. 78.8%; p < 0.001), ESR (61.6 vs. 86.8 mm/h; p < 0.001), and CRP level (24.4 vs. 84.8 mg/L; p < 0.001).

## Radiological findings

The radiological findings are summarized in Table 3. Thoracic vertebral involvement was observed more frequently in TS group than among those in PS group (57.5% vs. 23.5%; p < 0.001), whereas lumbar involvement was observed more frequently in PS group (75.0% vs. 53.4%; p = 0.008). The frequency of the following MRI findings was significantly higher in TS group than those in PS group: number of vertebrae involved ≥ 3 (42.5% vs. 20.6%; p = 0.005), paraspinal abscess (70.4% vs. 31.8%; p < 0.001), post-contrast well-defined margin enhancement (50.0% vs. 10.6%; p < 0.001), thin abscess wall (29.6% vs. 7.6%; p < 0.001) and anterior subligamentous abscess (81.9% vs. 57.4%; p = 0.002). There were no significant differences in the frequencies of skipping lesions, disc involvement, disc height loss, epidural abscess and posterior element involvement between groups.

## The biomarker cutoff values for differentiating TS from PS

The ROC curve with Youden index was used to determine the cutoff value of each biomarker, TS group was associated with the following: WBC count ≤ 9,700/mm$^3$, CRP level ≤ 50 mg/L, ESR ≤ 92 mm/h, and neutrophil fraction ≤ 78% (Table 4). The accuracy of the biomarkers for

**Table 3. Radiologic findings in tuberculous and pyogenic spondylodiscitis.**

| Variables | All (n = 141) | TS (n = 73) | PS (n = 68) | P value |
|---|---|---|---|---|
| Level of vertebrae involvement, n (%) | | | | |
| Cervical[†] | 16 (11.3) | 8 (11.0) | 8 (11.8) | 0.880 |
| Thoracic[†] | 58 (41.1) | 42 (57.5) | 16 (23.5) | <0.001[‡] |
| Lumbar[†] | 90 (63.8) | 39 (53.4) | 51 (75.0) | 0.008[‡] |
| Sacrum[†] | 7 (5.0) | 4 (5.5) | 3 (4.4) | 1.000 |
| No. of vertebrae involved | | | | 0.005[‡] |
| 1–2[†] | 96 (68.1) | 42 (57.5) | 54 (79.4) | |
| ≥ 3[†] | 45 (31.9) | 31 (42.5) | 14 (20.6) | |
| Skipping lesions[†] | 19 (13.5) | 11 (15.1) | 8 (11.8) | 0.566 |
| Disc involvement[†] | 118 (83.7) | 62 (84.9) | 56 (82.4) | 0.679 |
| Disc height loss[†] | 89 (63.1) | 44 (60.3) | 45 (66.2) | 0.468 |
| Epidural abscess[†] | 101 (72.1) | 56 (77.8) | 45 (66.2) | 0.126 |
| Paraspinal abscess[†] | 71 (51.8) | 50 (70.4) | 21 (31.8) | <0.001[‡] |
| Paraspinal soft tissue enhancement | | | | <0.001[‡] |
| Postcontrast well-defined margin[†] | 42 (30.9) | 35 (50.0) | 7 (10.6) | |
| Postcontrast ill-defined margin[†] | 51 (37.5) | 22 (31.4) | 29 (43.9) | |
| No paraspinal soft tissue enhance[†] | 43 (31.6) | 13 (18.6) | 30 (45.5) | |
| Wall thickness <0.001[‡] | | | | <0.001[‡] |
| Thin[†] | 26 (19.0) | 21 (29.6) | 5 (7.6) | |
| Thick[†] | 40 (29.2) | 26 (36.6) | 14 (21.2) | |
| No wall[†] | 71 (51.8) | 24 (33.8) | 47 (71.2) | |
| Anterior subligamentous abscess[†] | 98 (70.0) | 59 (81.9) | 39 (57.4) | 0.002[‡] |
| Posterior element involvement[†] | 64 (45.4) | 30 (41.1) | 34 (50.0) | 0.247 |

TS = Tuberculous spondylodiscitis, PS = pyogenic spondylodiscitis.

[†]The values are given as the number of patients, with the percentage in parentheses.

[‡]Significant ($p < 0.05$).

differentiating TS from PS, estimated with area under the curve (AUC) according to cutoff value of each biomarker, is shown in Fig 2.

## Predictive factors and scoring scheme for differentiating TS from PS

Using univariate analysis, factors associated with TS were determined from baseline patient demographics, clinical presentations, laboratory data, and radiological findings. The results of this analysis are presented in Table 4. Univariate analysis showed that 16 variables—age, history of diabetes, history of tuberculosis, time elapsed to diagnosis, fever more than 38°C, WBC count ≤ 9,700/mm³, CRP level ≤ 50 mg/L, ESR ≤ 92 mm/h, neutrophil fraction ≤ 78%, thoracic involvement, non-lumbar vertebral involvement, number of involved vertebrae, paraspinal abscess, post-contrast well-defined margin enhancement, abscess wall thickness and anterior subligamentous involvement—had a p value < 0.1 and were considered in the multiple logistic regression. The multivariate logistic regression model revealed that WBC ≤ 9,700/mm³ (odds ratio [OR] 13.11, 95% confidence interval [CI] 4.23–40.61), neutrophil fraction ≤ 78% (OR 4.93, 95% CI 1.59–15.30), ESR ≤ 92 mm/hr (OR 4.07, 95% CI 1.24–13.36) and presence of paraspinal abscess on MRI (OR 10.25, 95% CI 3.17–33.13) were significant predictors of TS (Table 5). The p value for the Hosmer-Lemeshow goodness-of-fit test was

**Table 4. Univariate analysis of potential predictors for tuberculous spondylodiscitis over pyogenic spondylodiscitis.**

| Predictor | Odds Ratio | 95% CI | P value |
|---|---|---|---|
| Age | 0.98 | 0.96–1.00 | 0.026[‡] |
| Diabetes mellitus = No | 2.35 | 1.03–5.37 | 0.043[‡] |
| Chronic liver disease = No | 3.44 | 0.67–17.64 | 0.139 |
| History of tuberculosis | 52.29 | 6.88–397.34 | <0.001[‡] |
| Time elapsed to diagnosis (days) | 1.01 | 1.00–1.02 | <0.001[‡] |
| Absence of fever (T less than 38˚C) | 5.52 | 2.62–11.64 | <0.001[‡] |
| WBC count<br>≤ 9700/mm$^3$<br>> 9700/mm$^3$ | 17.80<br>1 | 7.69–41.22 | < 0.001[‡] |
| CRP level (mg/L)<br>≤ 50<br>> 50 | 8.65<br>1 | 4.02–18.62 | < 0.001[‡] |
| ESR (mm/h)<br>≤ 92<br>> 92 | 8.62<br>1 | 3.59–20.67 | < 0.001[‡] |
| Neutrophil fraction<br>≤ 78%<br>> 78% | 9.95<br>1 | 4.48–22.07 | < 0.001[‡] |
| Thoracic vertebrae involvement | 4.40 | 2.13–9.11 | < 0.001[‡] |
| Lumbar vertebrae involvement = No | 2.62 | 1.28–5.35 | 0.009[‡] |
| Number of vertebrae involvement | 2.85 | 1.35–6.02 | 0.006[‡] |
| Skipping lesion | 1.33 | 0.50–3.54 | 0.567 |
| Paraspinal abscess | 5.10 | 2.47–10.55 | <0.001[‡] |
| Postcontrast well-defined signal margin | 11.54 | 4.08–32.65 | <0.001[‡] |
| Wall thickness | | | |
| Thin wall | 8.22 | 2.76–24.52 | <0.001[‡] |
| Thick wall | 3.64 | 1.61–8.21 | 0.002[‡] |
| Anterior subligamentous involvement | 3.37 | 1.56–7.28 | 0.002[‡] |

CI = confidence interval, T = temperature, WBC = white blood cell, ESR = erythrocyte sedimentation rate, CRP = C-reactive protein.

[‡]Factors with a p value less than or equal to 0.1 were considered to be potential predictors for tuberculous spondylodiscitis and were considered when building the multivariate logistic regression model.

0.107, which was greater than the significant threshold. Based on the ROC curve analysis, the AUC of this model was 0.921 (Fig 3).

The scoring scheme was developed, as displayed in Table 6, using the rounded up estimated OR of the four significant variables described in Table 5. These scores were used to stratify the probability of TS into three categories, which were low (score less than 13), moderate (score between 13 and 19) and high (score ≥ 19) probability; with TS prevalence of 8.1%, 29.6% and 82.2%, respectively. Illustrative examples using the scheme are shown in Figs 4 and 5.

## Discussion

Differentiating TS from PS is a cardinal diagnostic step in the treatment of SIS. Although previous reports have evaluated the clinical characteristics, laboratory data, and radiological findings of SIS originating from TS and PS, a scoring system aiming to identify TS among SIS patients has not been reported [3,11]. The present study identified four predictive factors— ESR, presence of paraspinal abscess on MRI, neutrophil fraction, and WBC count—as being helpful to distinguish TS from PS (Table 5) with excellent predictability (AUC = 0.921). This model was converted to a scoring scheme that justified an increased probability of TS among

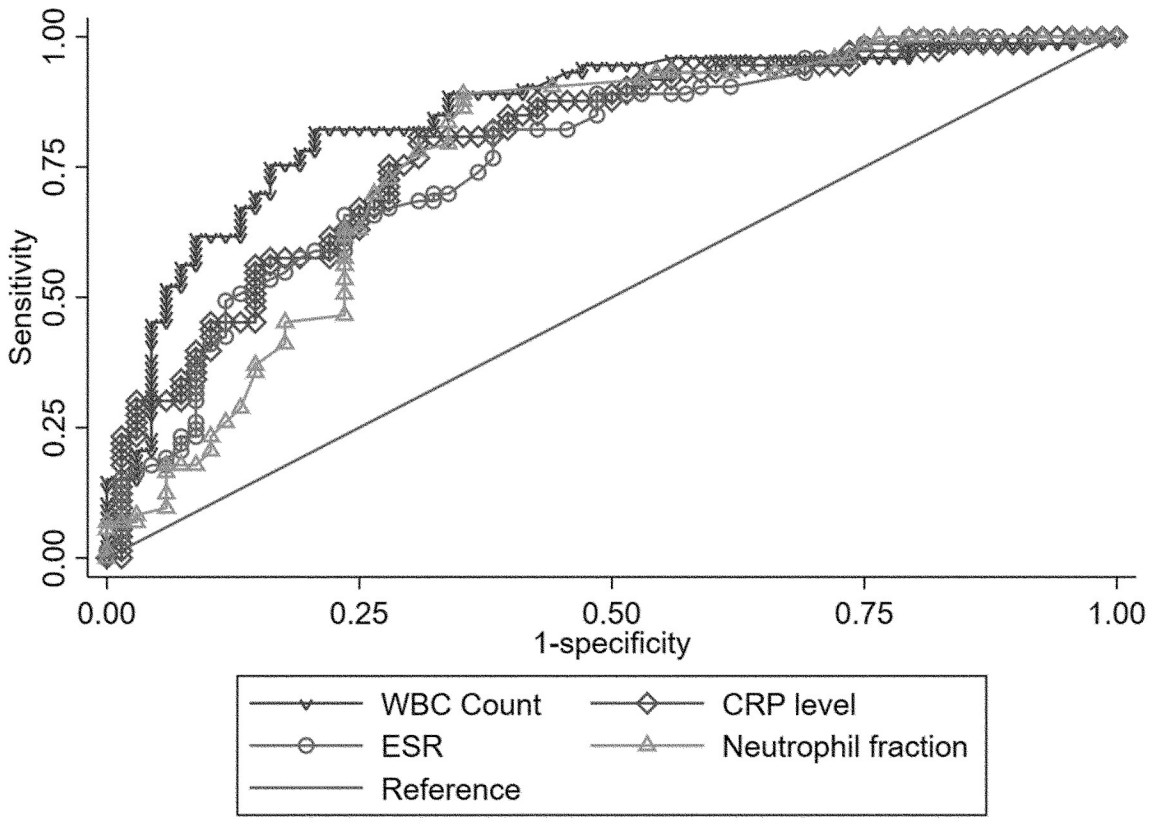

| Variables | Cut-off values | AUC (95% CI) | % | | | |
| --- | --- | --- | --- | --- | --- | --- |
| | | | Sensitivity | Specificity | PPV | NPV |
| WBC count | 9,700/mm³ | 0.81 (0.74, 0.87) | 82.2 | 79.4 | 81.1 | 80.6 |
| CRP level | 50 mg/L | 0.74 (0.67, 0.82) | 79.5 | 69.1 | 73.4 | 75.8 |
| ESR | 92 mm/h | 0.70 (0.63, 0.77) | 89.0 | 51.5 | 66.3 | 81.4 |
| Neutrophil fraction | 78% | 0.75 (0.68, 0.82) | 83.6 | 66.2 | 72.6 | 78.9 |

**Fig 2. Receiver-operating characteristic curve of white blood cell counts, C-reactive protein levels, erythrocyte sedimentation rate, and neutrophil fraction to differentiate tuberculous spondylodiscitis from pyogenic spondylodiscitis.** AUC = area under the curve, CI = confidence interval, PPV = positive predictive value, NPV = negative predictive value, WBC = white blood cell, ESR = erythrocyte sedimentation rate, CRP = C-reactive protein.

SIS patients (Table 6). Previous studies have reported a regression model for differentiate TS from PS but did not simplify this to a point score or provide any suggestions about interpretation of probabilities of TS [3,11]. To the best of our knowledge, this is the first study to date to develop the scoring system to assist physicians with decision-making to diagnose TS among SIS patients.

**Table 5. Results of multivariate logistic regression of independent predictors of tuberculous spondylodiscitis.**

| Predictor | Odds Ratio | 95% CI | P value |
|---|---|---|---|
| WBC count | | | |
| $\leq 9700/mm^3$ | 13.11 | 4.23–40.61 | <0.001[‡] |
| $> 9700/mm^3$ | 1 | | |
| ESR (mm/h) | | | |
| $\leq 92$ | 4.07 | 1.24–13.36 | 0.020[‡] |
| $> 92$ | 1 | | |
| Neutrophil fraction | | | |
| $\leq 78\%$ | 4.93 | 1.59–15.30 | 0.006[‡] |
| $> 78\%$ | 1 | | |
| Paraspinal abscess | | | |
| Yes | 10.25 | 3.17–33.13 | <0.001[‡] |
| No | 1 | | |

CI = confidence interval, WBC = white blood cell, ESR = erythrocyte sedimentation rate.

[‡]Significant ($p < 0.05$).

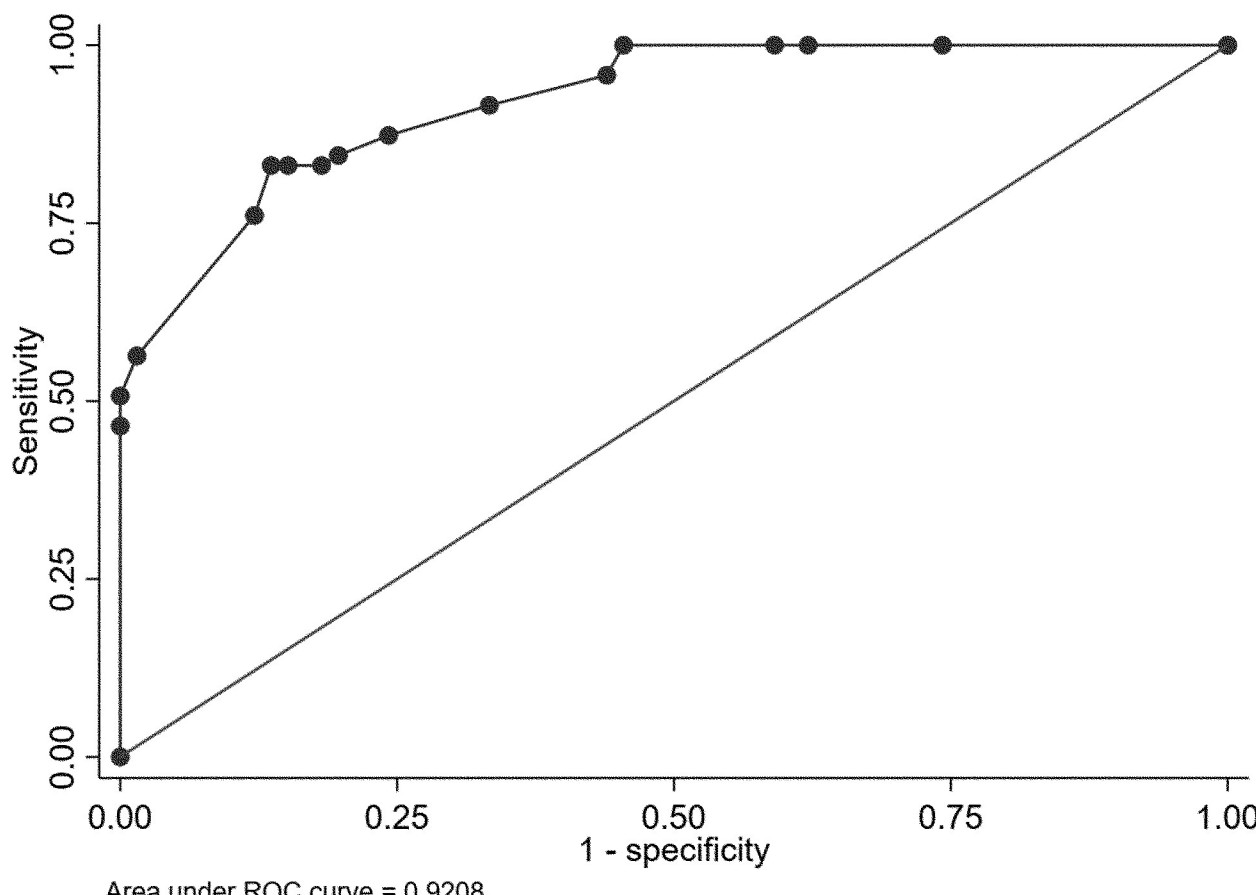

Area under ROC curve = 0.9208

**Fig 3. Receiver-operating characteristic curve for tuberculous spondylodiscitis using the predictive probability of a multivariate logistic regression model.**

**Table 6. Scoring scheme of prediction model to differentiate tuberculous spondylodiscitis from pyogenic spontaneous spondylodiscitis.**

| Factors | Scoring | Score for individual |
|---|---|---|
| **WBC Group** | | |
| WBC count $\leq$ 9,700/mm$^3$ | 13 | |
| WBC count > 9,700/mm$^3$ | 1 | |
| **ESR Group** | | |
| ESR $\leq$ 92 mm/hr | 4 | |
| ESR > 92 mm/hr | 1 | |
| **Neutrophil fraction Group** | | |
| Neutrophil fraction $\leq$ 78% | 5 | |
| Neutrophil fraction > 78% | 1 | |
| **Presence of paraspinal abscess in MRI** | | |
| Yes | 10 | |
| No | 1 | |
| **Total score** | Low probability for TS | less than13 |
| (Sum of each significant factor) | Moderate probability for TS | 13 to <19 |
| | High probability for TS | $\geq$ 19 |

TS = tuberculous spondylodiscitis, WBC = white blood cell, ESR = erythrocyte sedimentation rate, MRI = magnetic resonance imaging.

The present results indicate several potentially useful distinguishing features. Back pain, which represented the majority presenting symptoms, was not significantly different between groups. In contrast, absence of fever and a longer time elapsed to diagnosis were more associated with TS (Table 2) than with PS. Yoon et al. reported that fever was less common symptom in TS compared to PS patients, median time elapsed to diagnosis of spondylodiscitis more than 7 days was a risk factor associated with TS, and bacteremia was significantly associated with PS [11]. We found 64.7% of patients with PS had previous or active bacteremia at time of diagnosis, whereas no patients with TS had bacteremia ($P < 0.001$). We suggest that, if SIS is suspected and patients have medically unstable circumstance (e.g., severe sepsis), then antibiotic specified to identified organism from blood culture, should be considered [20].

Inflammatory markers—such as ESR, WBC, neutrophil count and CRP level—are highly sensitive in the diagnosis of bone and joint infection [21–23]. Using the cutoff value by the ROC curve with Youden index, our results suggest that ESR $\leq$ 92 mm/hr, neutrophil fraction $\leq$ 78% and WBC $\leq$ 9,700/mm$^3$, are highly suggestive diagnostic cluesfor differentiating patients with TS from those with PS (Table 5). Our findings were consistent with prior reports. Kim et al. reported the following biomarker cutoff values indicating TS: WBC count: 10,000/mm$^3$, neutrophil fraction $\leq$ 75%, ESR: 40 mm/hr, CRP more than 5 mg/dL, and alkaline phosphatase less than 120 IU/L [3]. Yoon et al. reported the following biomarker cutoff values to differentiate TS from PS: WBC count: 9,130 mm$^3$, CRP 7.37 mg/dL, and procalcitonin level: 0.11 ng/dL [11]. However, the results of the multivariate logistic regression model in our study did not indicate CRP as a predictive factor. This finding might be partly explained by the nature of our patient cohort. As shown in our study, the median elapsed time to diagnosis in both groups was delayed, so CRP level may not have been more predominant than ESR level at the time of diagnosis [21,22].

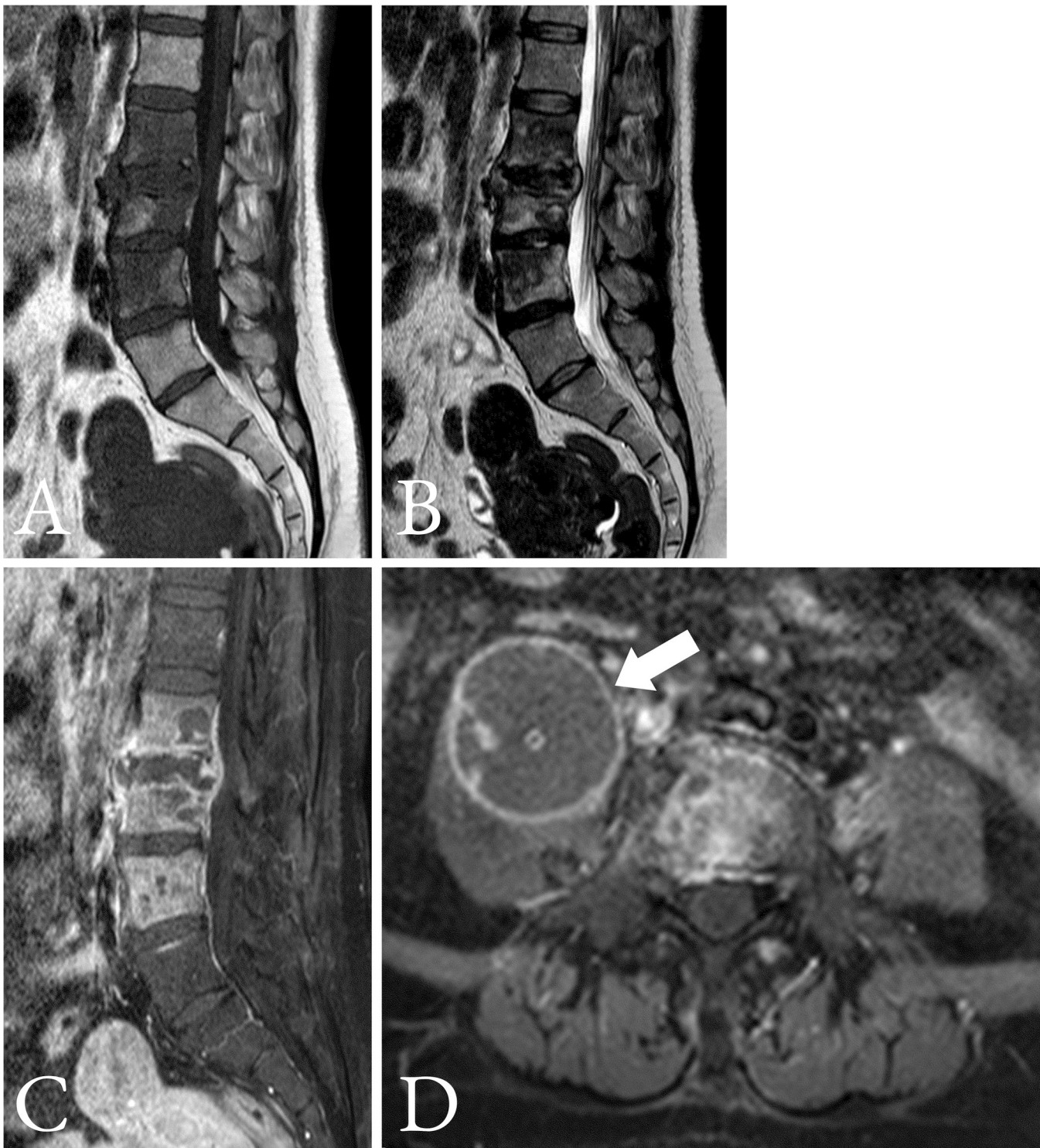

**Fig 4. A 53-year-old woman with infectious spondylodiscitis. a)** Sagittal T1-weighted MRI scan demonstrates L2–L4 hypointensity with anterior subligamentous spreading at L2–L3. **b)** Sagittal T2-weighted MRI scan demonstrates L2–L4 inhomogeneous hyperintensity. **c)** Sagittal T1-weighted gadolinium-enhanced MRI scan demonstrates focal inhomogeneous at L2–L3 and intraosseous rim enhancement. **d)** Axial T1-weighted gadolinium-enhanced MRI scan demonstrates a well-defined paraspinal abscess (white arrow) at L3 level. The patient's laboratory data were as follows: WBC $\leq$ 9,700/mm$^3$, neutrophil fraction $\leq$ 78%, and ESR $\leq$ 92 mm/h. Using the scoring scheme, the patient's score was 10 + 13 + 5 + 4 = 32; indicating a high probability of tuberculous spondylodiscitis. MRI = magnetic resonance imaging, WBC = white blood cell, ESR = erythrocyte sedimentation rate.

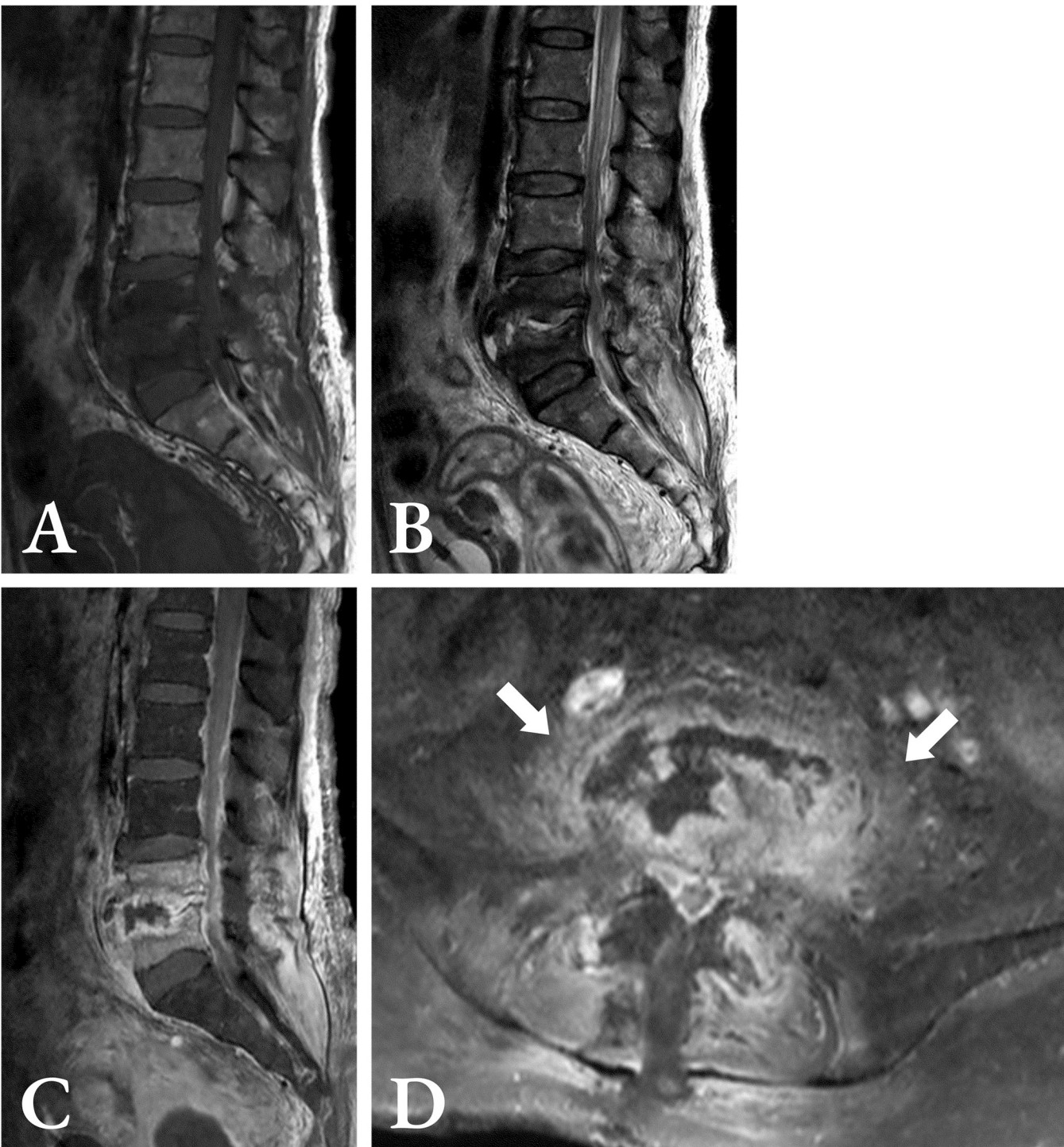

**Fig 5. A 62-year-old woman with infectious spondylodiscitis. a)** Sagittal T1-weighted MRI scan demonstrates hypointensity of L4-L5 vertebral body. **b)** Sagittal T2-weighted MRI scan demonstrates isointensity of the vertebral bodies and destruction of the intervertebral disc. **c)** Sagittal T1-weighted gadolinium-enhanced MRI scan demonstrates homogeneous enhancement of the vertebral bodies and a disc abscess with peridiscal rim enhancement. d) Axial T1-weighted gadolinium-enhanced MRI scan at disc level demonstrates rim enhancement of a peridiscal abscess and no obvious paraspinal abscess (white arrow). The patient's laboratory data were as follows: WBC more than 9,700/mm$^3$, neutrophil fraction more than 78%, and ESR more than 92 mm/hr. Using the scoring scheme, this patient's score was 1 + 1 + 1 + 1 = 4; indicating a low probability of tuberculous spondylodiscitis. MRI = magnetic resonance imaging, WBC = white blood cell, ESR = erythrocyte sedimentation rate.

MRI is the imaging modality of choice for evaluation of spinal infection, with good sensitivity, specificity, and accuracy [24,25]. Previous report indicated that contrast-enhanced MRI is robust technique for differentiating TS from PS [18]. Our study demonstrated a presence of paraspinal abscess (with post-contrast well-defined margin) as a predictor for differentiating TS from PS (Tables 4 and 5). Supporting this finding, a previous report indicated that a well-defined paraspinal abscess was one of the most distinctive findings of TS while PS tended to show more diffuse, ill-defined areas of enhancement [18]. Recently, Kanna et al. also demonstrated that large abscess with thin wall is one of the MRI findings those were highly suggestive of TS [26].

Tuberculosis remains a global problem; while the highest incidence appears in developing countries, tuberculosis is also increasingly prevalent in developed countries. The Southeast Asian region alone accounted for 44% of the global burden [27]. In tuberculosis-endemic areas, an early diagnosis of TS is paramount for appropriate treatment. Previous reports demonstrated risk factors for TS that included absence of fever, constitutional symptoms, neurological deficit, active tuberculosis other than spine, longer elapsed time to diagnosis, presence of paraspinal abscess, increasing ESR, WBC count $\leq 9,600/mm^3$, and neutrophil fraction $\leq 75\%$ [3,10,11,28]. The fact that these four predictive factors in our study did not differ from previous findings reflects that the factors are robust and enhance the generalizability of this predictive model. Therefore, if the causative organisms remain unidentified in SIS patients, we recommend comprehensive review of the clinical history, inflammatory marker profiles, and MRI findings. Patients with the predictive factors for TS, as demonstrated in our study, should be considered for tuberculous infection particularly for an endemic area.

The clinical prediction model of this study had a number of strengths. We included all parameters relevant to the diagnosis in the model and converted the regression equation into a scoring scheme for clinical use. This scheme provides useful information for differentiating TS from PS patients. However, there were some limitations in this study. First, it was a single-center retrospective analysis. Second, our center is a referral center, which may have skewed the clinical or laboratory data regarding the onset of symptoms. Studies with a larger population are required to evaluate the clinical relevance of our results. Third, this study comprised by data for two cohorts collected in different time period i.e. cohort year 2016 (January 2008 to December 2016) and cohort year 2021 (January 2017 to December 2021) using the same protocol. This approach may have involved bias owing to trends in spondylodiscitis management over the 13-year period. However, we believe that the extended cohort, would enhance the validity of the study and made the results more precise for use in current practice. The clinical predictive model with the scoring scheme was ultimately shown to be an accurate tool to predict the probability of TS, among SIS patients, to assist physicians with decision-making. However, external validity of this scoring scheme still needs to be confirmed. Although not the objective of the study, the clinical presentations of other low virulent causative organisms, i.e. mycobacterium other than TB, fungi or brucellosis, may mimic TS which need to be clarified in future studies. Finally, we did not include the patients who had concomitant TS and PS in our study (Fig 1) so this may be limit the generalizability of our results for such patients who had concomitant TS and PS.

## Conclusions

This prediction model incorporating WBC counts, neutrophil fraction, ESR, and the presence of paraspinal abscess accurately predicted the causative pathogens of spondylodiscitis. The novel scoring scheme with combination of these biomarkers and radiological features can be

useful to differentiate TS from PS. Further investigation with a larger sample size is warranted for clarification.

## Supporting information

**S1 Checklist. STROBE statement—checklist of items that should be included in reports of observational studies.**
(DOCX)

**S1 File.**
(XLSX)

## Acknowledgments

The authors wish to thank Assoc. Prof. Dr. Paphon Sa-Ngasoongsong, Department of Orthopaedics, Faculty of Medicine Ramathibodi Hospital, Mahidol University, Bangkok, Thailand for editing the manuscript.

## Author Contributions

**Conceptualization:** Thamrong Lertudomphonwanit, Siriorn Watcharananan, Pongsthorn Chanplakorn.

**Data curation:** Thamrong Lertudomphonwanit, Chirtwut Somboonprasert, Kittiphon Lilakhunakon, Pongsthorn Chanplakorn.

**Formal analysis:** Thamrong Lertudomphonwanit, Sasivimol Rattanasiri, Pongsthorn Chanplakorn.

**Investigation:** Suphaneewan Jaovisidha, Thumanoon Ruangchaijatuporn, Praman Fuangfa, Pongsthorn Chanplakorn.

**Methodology:** Siriorn Watcharananan, Pongsthorn Chanplakorn.

**Supervision:** Suphaneewan Jaovisidha, Pongsthorn Chanplakorn.

**Writing – original draft:** Thamrong Lertudomphonwanit, Pongsthorn Chanplakorn.

**Writing – review & editing:** Thamrong Lertudomphonwanit, Pongsthorn Chanplakorn.

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
