## [Decision Letter · Decision Letter 0]

3 Jul 2023

PONE-D-23-17746A Clinical Prediction Model to Differentiate Tuberculous Spondylodiscitis from Pyogenic Spontaneous SpondylodiscitisPLOS ONE

Dear Dr. Chanplakorn,

Thank you for submitting your manuscript to PLOS ONE. After careful consideration, we feel that it has merit but does not fully meet PLOS ONE’s publication criteria as it currently stands. Therefore, we invite you to submit a revised version of the manuscript that addresses the points raised during the review process.

We look forward to receiving your revised manuscript.

Kind regards,

Mohamed El-Sayed Abdel-Wanis, Ph.D.

Academic Editor

PLOS ONE

Journal Requirements:

Additional Editor Comments:

This is a well-written manuscript. The authors make good use of figures and tables to highlight the literature search and data obtained from the analysis. The discussion is robust and well-cited.

Introduction

. Lines 59-60: “The incidence of SIS ranges from 8 to 24 cases/million inhabitants/year” Is this a global incidence or in in Thailand?  

. Materials and Methods  

. Line 109: “History of bacteremia or sepsis…” I think better to be corrected to “History suggestive of bacteremia or sepsis…” the authors must mention also some of the symptoms that they considered suggestive of bacteremia or sepsis.  

. Results

Line 153: “..history of bacteremia..” should be corrected to “..“history suggestive of bacteremia”

Discussion

. Line 293: “…then empirical antibiotic specified  to identified organism from blood culture, should be considered [20].”  As antibiotics is given according to culture, it could not be considered empirical. So, “empirical” should be deleted.

. Line 298: “…are diagnostic clues..” better to be changed to “.. are highly suggestive diagnostic clues..” 

. Lines 319-321: “This study identified ESR, neutrophil  fraction, WBC count and presence of paraspinal abscess on MRI as predictive factors for  differentiating TS from PS”. The data in this sentence was previously reported in the “discussion and this sentence should be deleted.  

. Lines 341- 343: “However, we believe that the extended cohort, which included more patients compared with not using the extended cohort, would enhance the validity of the study and made the results more precise for use in current practice.”  The sentence should be better corrected to: “However, we believe that the extended cohort, would enhance the validity of the study and made the results more precise for use in current practice.”

Reviewers' comments:

Reviewer's Responses to Questions

**Comments to the Author**

1. Is the manuscript technically sound, and do the data support the conclusions?

Reviewer #1: Yes

Reviewer #2: Yes

Reviewer #3: Yes

2. Has the statistical analysis been performed appropriately and rigorously? 

Reviewer #1: Yes

Reviewer #2: I Don't Know

Reviewer #3: Yes

3. Have the authors made all data underlying the findings in their manuscript fully available?

Reviewer #1: Yes

Reviewer #2: Yes

Reviewer #3: Yes

4. Is the manuscript presented in an intelligible fashion and written in standard English?

Reviewer #1: Yes

Reviewer #2: Yes

Reviewer #3: Yes

5. Review Comments to the Author

Reviewer #1: I have read this very interesting manuscript, and have only one comment:

How did the authors account/differentiate patients who had concomitant TS and PS. There are several reports in the literature of combined tuberculous and pyogenic discitis and I would like the authors to elaborate on that.

Reviewer #2: New idea. Of course its evidence base. Somehow its contrary to our clinical practice. We see high ESR level in Tuberculosis patient as compared to bacterial patient, probably due to delay in diagnosis of TB Spine patient in our community. Due to rapid rise in Blood markers in Bacterial spine infection it picked earlier.

> and < these symbols are confusing as which one represent less than and which one more than. Therefore I suggest somewhere in article this must be mentioned in words that < means less than and > means more than.

Reviewer #3: Thank you for your submission and the very good work.

Abstract: well structured and clearly describing the contents of the manuscript

Methodology: Please add more details to the description of the study setting including the bed capacity. Government VS Private facility, Type of reimbursement system whether it is cash vs insurance based

Results: The tables are clear. Applying regression analysis was very important in such studies to accommodate for confounding factors.

Discussion: Very smooth flow of the literature review highlighting some of the mot important publications in the same field and the same pathology

6. PLOS authors have the option to publish the peer review history of their article (what does this mean?). If published, this will include your full peer review and any attached files.

Reviewer #1: No

Reviewer #2: No

Reviewer #3: No

---

## [Author Response · Author response to Decision Letter 0]

26 Jul 2023

Response to Reviewer’s Comments

Response to Editor Comments

The authors would like to thank the editor for your very valuable comment. We have addressed every single issue that you raised as below.

1. Introduction: 

- Lines 59-60: “The incidence of SIS ranges from 8 to 24 cases/million inhabitants/year” Is this a global incidence or in in Thailand?

 Response: This is a global incidence.

 We have added the below statements in Introduction section (page 2, line 33). 

 The global incidence of SIS ranges from 8 to 24 cases/million inhabitants/year [1,2].

2. Materials and Methods: Line 109: “History of bacteremia or sepsis…” I think better to be corrected to “History suggestive of bacteremia or sepsis…” the authors must mention also some of the symptoms that they considered suggestive of bacteremia or sepsis.

 Response: We have corrected “History of bacteremia or sepsis” to “History suggestive of bacteremia or sepsis” and added some of the symptoms of bacteremia as you suggested. (page 4, line 89)

3. Results

- Line 153: “..history of bacteremia..” should be corrected to “..“history suggestive of bacteremia”

Response: We have corrected “history of bacteremia” to “history suggestive of bacteremia” as you suggested. (page 6 , line 135)

4. Discussion

- Line 293: “…then empirical antibiotic specified to identified organism from blood culture, should be considered [20].” As antibiotics is given according to culture, it could not be considered empirical. So, “empirical” should be deleted.

Response: We have deleted “empirical” as you suggested. (page 15, line 276)

- Line 298: “…are diagnostic clues..” better to be changed to “.. are highly suggestive diagnostic clues..”

Response: We have corrected “..are diagnostic clues..” to “.. are highly suggestive diagnostic clues..” as you suggested. (page 15, line 281)

- Lines 319-321: “This study identified ESR, neutrophil fraction, WBC count and presence of paraspinal abscess on MRI as predictive factors for differentiating TS from PS”. The data in this sentence was previously reported in the “discussion and this sentence should be deleted.

Response: We have deleted “This study identified ESR, neutrophil fraction, WBC count and presence of paraspinal abscess on MRI as predictive factors for differentiating TS from PS” as you suggested. (page 16, line 302) 

- Lines 341- 343: “However, we believe that the extended cohort, which included more patients compared with not using the extended cohort, would enhance the validity of the study and made the results more precise for use in current practice.” The sentence should be better corrected to: “However, we believe that the extended cohort, would enhance the validity of the study and made the results more precise for use in current practice.”

 Response: We have corrected “However, we believe that the extended cohort, which included more patients compared with not using the extended cohort, would enhance the validity of the study and made the results more precise for use in current practice.” to: “However, we believe that the extended cohort, would enhance the validity of the study and made the results more precise for use in current practice.” as you suggested. (page 17, line 324)

Reviewer #1: I have read this very interesting manuscript, and have only one comment:

How did the authors account/differentiate patients who had concomitant TS and PS. There are several reports in the literature of combined tuberculous and pyogenic discitis and I would like the authors to elaborate on that.

Response to Reviewer # 1:

We appreciate your valuable comment. There were no patients who had concomitant tuberculous and pyogenic discitis in our cohort. We excluded two patients with simultaneous pyogenic spondylodiscitis and tuberculous spondylodiscitis (Figure 1). 

We have added the below statements in Materials and Methods section (exclusion criteria) (page 4, line 81). 

“…and (4) patients with simultaneous PS and TS (Fig 1).” 

We also added the below statements in Discussion section.

Finally, we did not include the patients who had concomitant TS and PS in our study (Fig 1) so this may be limit the generalizability of our results for such patients who had concomitant TS and PS. (page 18, line 333).

Reviewer #2: New idea. Of course its evidence base. Somehow its contrary to our clinical practice. We see high ESR level in Tuberculosis patient as compared to bacterial patient, probably due to delay in diagnosis of TB Spine patient in our community. Due to rapid rise in Blood markers in Bacterial spine infection it picked earlier.

> and < these symbols are confusing as which one represent less than and which one more than. Therefore I suggest somewhere in article this must be mentioned in words that < means less than and > means more than.

Response to Reviewer # 2:

The authors would like to thank the reviewer for your valuable comment.

We have corrected ”>” to “more than” and “<” to “less than” as you suggested as below.

In Table 4: Absence of fever (T less than 38°C) (page 11)

fever more than 38°C (Line 206, page 12)

These scores were used to stratify the probability of TS into three categories, which were low (score less than 13) (Line 227, page 13)

In table 6: score less than 13 (page 13)

Fig. 5 Caption: The patient’s laboratory data were as follows: WBC more than 9,700/mm3, neutrophil fraction more than 78%, and ESR more than 92 mm/hr. (Line 252, page 14)

Yoon et al. reported that fever was less common symptom in TS compared to PS patients, median time elapsed to diagnosis of spondylodiscitis more than 7 days was a risk factor associated with TS, and bacteremia was significantly associated with PS [11] (Line 272, page 15)

Kim et al. reported the following biomarker cutoff values indicating TS: WBC count: 10,000/mm3, neutrophil fraction ≤ 75%, ESR: 40 mm/hr, CRP more than 5 mg/dL, and alkaline phosphatase less than 120 IU/L [3]. (Line 284, page 16)

Reviewer #3: Thank you for your submission and the very good work.

Abstract: well structured and clearly describing the contents of the manuscript

Methodology: Please add more details to the description of the study setting including the bed capacity. Government VS Private facility, Type of reimbursement system whether it is cash vs insurance based

Results: The tables are clear. Applying regression analysis was very important in such studies to accommodate for confounding factors.

Discussion: Very smooth flow of the literature review highlighting some of the mot important publications in the same field and the same pathology

Response to Reviewer # 3:

The authors would like to thank the reviewer for your valuable comment. We have addressed the issue that you raised as below.

- Methodology: Please add more details to the description of the study setting including the bed capacity. Government VS Private facility, Type of reimbursement system whether it is cash vs insurance based

Response: Our hospital is 1300-bed, University hospital with government support. Reimbursement systems at our hospital are covered by multiple sources such as cash, universal coverage by government, workers’ insurance, government pay for government personnel, and private insurance.

We have added the above statements in Materials and Methods section (page 3-4, line 67-71).

The edited parts were highlighted with track changes. 

Thank you very much for your help in improving this manuscript.

---

## [Editor Report · Decision Letter 1]

31 Jul 2023

PONE-D-23-17746R1A Clinical Prediction Model to Differentiate Tuberculous Spondylodiscitis from Pyogenic Spontaneous SpondylodiscitisPLOS ONE

Dear Dr. Chanplakorn,

Thank you again for addressing the editorial  and reviewer comments. I would like to recommend minimal correction as follow:

Data collection Lines 89-91

.                 History suggestive of bacteremia (the presence of viable bacteria in the circulating  blood with or without following symptoms: fever, chill, tachypnea, altered sensorium or hypotension, etc.)  should be corrected to:   “History suggestive of bacteremia e.g. fever and chill.”

We look forward to receiving your revised manuscript.

Kind regards,

Mohamed El-Sayed Abdel-Wanis, Ph.D.

Academic Editor

PLOS ONE
---

## [Author Response · Author response to Decision Letter 1]

2 Aug 2023

Response to Reviewer’s Comments

Response to Editor Comments

The authors would like to thank the editor for your very valuable comment. We have addressed the issue that you raised below.

Editor’s comment

Data collection Lines 89-91

1. History suggestive of bacteremia (the presence of viable bacteria in the circulating blood with or without following symptoms: fever, chill, tachypnea, altered sensorium or hypotension, etc.) should be corrected to: “History suggestive of bacteremia e.g. fever and chill.”

History suggestive of bacteremia e.g. fever and chill.

Response: We have corrected “History suggestive of bacteremia (the presence of viable bacteria in the circulating blood with or without following symptoms: fever, chill, tachypnea, altered sensorium or hypotension, etc.)” to “History suggestive of bacteremia e.g. fever and chill.” as you suggested. (Page 4 line 89)

It was highlight in gray. 

Thank you very much for your help in improving this manuscript.

---

## [Editor Report · Decision Letter 2]

7 Aug 2023

A Clinical Prediction Model to Differentiate Tuberculous Spondylodiscitis from Pyogenic Spontaneous Spondylodiscitis

PONE-D-23-17746R2

Dear Dr. Pongsthorn Chanplakorn

We’re pleased to inform you that your manuscript has been judged scientifically suitable for publication and will be formally accepted for publication once it meets all outstanding technical requirements.

Kind regards,

Mohamed El-Sayed Abdel-Wanis, Ph.D.

Academic Editor

PLOS ONE

---

## [Editor Report · Acceptance letter]

10 Aug 2023

PONE-D-23-17746R2 

A Clinical Prediction Model to Differentiate Tuberculous Spondylodiscitis from Pyogenic Spontaneous Spondylodiscitis 

Dear Dr. Chanplakorn:

I'm pleased to inform you that your manuscript has been deemed suitable for publication in PLOS ONE. Congratulations! Your manuscript is now with our production department. 

Kind regards, 

on behalf of

Prof. Dr Mohamed El-Sayed Abdel-Wanis 

Academic Editor

PLOS ONE